# TreeQN and ATreeC:
# Differentiable Tree-Structured Models for Deep Reinforcement Learning

**Gregory Farquhar**[1]
gregory.farquhar@cs.ox.ac.uk

**Tim Rocktäschel**[1]
tim.rocktaschel@cs.ox.ac.uk

**Maximilian Igl**[1]
maximilian.igl@cs.ox.ac.uk

**Shimon Whiteson**[1]
shimon.whiteson@cs.ox.ac.uk

[1]University of Oxford, United Kingdom

## Abstract

Combining deep model-free reinforcement learning with on-line planning is a promising approach to building on the successes of deep RL. On-line planning with look-ahead trees has proven successful in environments where transition models are known a priori. However, in complex environments where transition models need to be learned from data, the deficiencies of learned models have limited their utility for planning. To address these challenges, we propose TreeQN, a differentiable, recursive, tree-structured model that serves as a drop-in replacement for any value function network in deep RL with discrete actions. TreeQN dynamically constructs a tree by recursively applying a transition model in a learned abstract state space and then aggregating predicted rewards and state-values using a tree backup to estimate $Q$-values. We also propose ATreeC, an actor-critic variant that augments TreeQN with a softmax layer to form a stochastic policy network. Both approaches are trained end-to-end, such that the learned model is optimised for its actual use in the tree. We show that TreeQN and ATreeC outperform $n$-step DQN and A2C on a box-pushing task, as well as $n$-step DQN and *value prediction networks* (Oh et al., 2017) on multiple Atari games. Furthermore, we present ablation studies that demonstrate the effect of different auxiliary losses on learning transition models.

## 1 Introduction

A promising approach to improving model-free deep reinforcement learning (RL) is to combine it with on-line planning. The model-free value function can be viewed as a rough global estimate which is then locally refined on the fly for the current state by the on-line planner. Crucially, this does not require new samples from the environment but only additional computation, which is often available.

One strategy for on-line planning is to use look-ahead tree search (Knuth & Moore, 1975; Browne et al., 2012). Traditionally, such methods have been limited to domains where perfect environment simulators are available, such as board or card games (Coulom, 2006; Sturtevant, 2008). However, in general, models for complex environments with high dimensional observation spaces and complex dynamics must be learned from agent experience. Unfortunately, to date, it has proven difficult to learn models for such domains with sufficient fidelity to realise the benefits of look-ahead planning (Oh et al., 2015; Talvitie, 2017).

A simple approach to learning environment models is to maximise a similarity metric between model predictions and ground truth in the observation space. This approach has been applied with some success in cases where model fidelity is less important, *e.g.*, for improving exploration (Chiappa et al., 2017; Oh et al., 2015). However, this objective causes significant model capacity to be devoted to predicting irrelevant aspects of the environment dynamics, such as noisy backgrounds, at the expense of value-critical features that may occupy only a small part of the observation space (Pathak et al.,

2017). Consequently, current state-of-the-art models still accumulate errors too rapidly to be used for look-ahead planning in complex environments.

Another strategy is to train a model such that, when it is used to predict a value function, the error in those predictions is minimised. Doing so can encourage the model to focus on features of the observations that are relevant for the control task. An example is the *predictron* (Silver et al., 2017b), where the model is used to aid policy evaluation without addressing control. *Value prediction networks* (VPNs, Oh et al., 2017) take a similar approach but use the model to construct a look-ahead tree only when constructing bootstrap targets and selecting actions, similarly to *TD-search* (Silver et al., 2012). Crucially, the model is not embedded in a planning algorithm during optimisation.

We propose a new tree-structured neural network architecture to address the aforementioned problems. By formulating the tree look-ahead in a differentiable way and integrating it directly into the $Q$-function or policy, we train the entire agent, including its learned transition model, end-to-end. This ensures that the model is optimised for the correct goal and is suitable for on-line planning during execution of the policy.

Since the transition model is only weakly grounded in the actual environment, our approach can alternatively be viewed as a model-free method in which the fully connected layers of DQN are replaced by a recursive network that applies transition functions with shared parameters at each tree node expansion.

The resulting architecture, which we call TreeQN, encodes an inductive bias based on the prior knowledge that the environment is a stationary Markov process, which facilitates faster learning of better policies. We also present an actor-critic variant, ATreeC, in which the tree is augmented with a softmax layer and used as a policy network.

We show that TreeQN and ATreeC outperform their DQN-based counterparts in a box-pushing domain and a suite of Atari games, with deeper trees often outperforming shallower trees, and TreeQN outperforming VPN (Oh et al., 2017) on most Atari games. We also present ablation studies investigating various auxiliary losses for grounding the transition model more strongly in the environment, which could improve performance as well as lead to interpretable internal plans. While we show that grounding the reward function is valuable, we conclude that how to learn strongly grounded transition models and generate reliably interpretable plans without compromising performance remains an open research question.

## 2 BACKGROUND

We consider an agent learning to act in a Markov Decision Process (MDP), with the goal of maximising its expected discounted sum of rewards $R_t = \sum_{t=0}^{\infty} \gamma^t r_t$, by learning a policy $\pi(\mathbf{s})$ that maps states $\mathbf{s} \in \mathcal{S}$ to actions $a \in \mathcal{A}$. The state-action value function ($Q$-function) is defined as $Q^\pi(\mathbf{s}, a) = \mathbb{E}_\pi [R_t | \mathbf{s}_t = \mathbf{s}, a_t = a]$; the optimal $Q$-function is $Q^*(\mathbf{s}, a) = \max_\pi Q^\pi(\mathbf{s}, a)$.

The Bellman optimality equation writes $Q^*$ recursively as

$$Q^*(\mathbf{s}, a) = \mathcal{T}Q^*(\mathbf{s}, a) \equiv r(\mathbf{s}, a) + \gamma \sum_{\mathbf{s}'} P(\mathbf{s}'|\mathbf{s}, a) \max_{a'} Q^*(\mathbf{s}', a'),$$

where $P$ is the MDP state transition function and $r$ is a reward function, which for simplicity we assume to be deterministic. $Q$-learning (Watkins & Dayan, 1992) uses a single-sample approximation of the contraction operator $\mathcal{T}$ to iteratively improve an estimate of $Q^*$.

In deep $Q$-learning (Mnih et al., 2015), $Q$ is represented by a deep neural network with parameters $\theta$, and is improved by regressing $Q(\mathbf{s}, a)$ to a target $r + \gamma \max_{a'} Q(\mathbf{s}', a'; \theta^-)$, where $\theta^-$ are the parameters of a target network periodically copied from $\theta$.

We use a version of $n$-step $Q$-learning (Mnih et al., 2016) with synchronous environment threads. In particular, starting at a timestep $t$, we roll forward $n_{\text{env}} = 16$ threads for $n = 5$ timesteps each. We then bootstrap off the final states only and gather all $n_{\text{env}} \times n = 80$ transitions in a single batch for

the backward pass, minimising the loss:

$$\mathcal{L}_{\text{nstep-Q}} = \sum_{\text{envs}} \sum_{j=1}^{n} \left( \sum_{k=1}^{j} \left[ \gamma^{j-k} r_{t+n-k} \right] + \gamma^{j} \max_{a'} Q \left( \mathbf{s}_{t+n}, a', \theta^{-} \right) - Q \left( \mathbf{s}_{t+n-j}, a_{t+n-j}, \theta \right) \right)^{2}. \quad (1)$$

If the episode terminates, we use the remaining episode return as the target, without bootstrapping.

This algorithm's actor-critic counterpart is A2C, a synchronous variant of A3C (Mnih et al., 2016) in which a policy $\pi$ and state-value function $V(s)$ are trained using the gradient:

$$\Delta \theta = \sum_{\text{envs}} \sum_{j=1}^{n} \nabla_{\theta_{\pi}} \log \pi(a_{t+n-j}|s_{t+n-j}) A_j(s_{t+n-j}, a_{t+n-j}) + \beta \nabla_{\theta_{\pi}} H(\pi(s_{t+n-j}))$$
$$+ \alpha \nabla_{\theta_V} A_j(s_{t+n-j}, a_{t+n-j})^2, \quad (2)$$

where $A_j$ is an advantage estimate given by $\sum_{k=1}^{j} \gamma^{j-k} r_{t+n-k} + \gamma^{j} V(\mathbf{s}_{t+n}) - V(\mathbf{s}_{t+n-j})$, $H$ is the policy entropy, $\beta$ is a hyperparameter tuning the degree of entropy regularisation, and $\alpha$ is a hyperparameter controlling the relative learning rates of actor and critic.

These algorithms were chosen for their simplicity and reasonable wallclock speeds, but TreeQN can also be used in other algorithms, as described in Section 3. Our implementations are based on OpenAI Baselines (Hesse et al., 2017).

The canonical neural network architecture in deep RL with visual observations has a series of convolutional layers followed by two fully connected layers, where the final layer produces one output for each action-value. We can think of this network as first calculating an encoding $\mathbf{z}_t$ of the state $\mathbf{s}_t$ which is then evaluated by the final layer to estimate $Q^*(\mathbf{s}_t, a)$ (see Fig. 1).

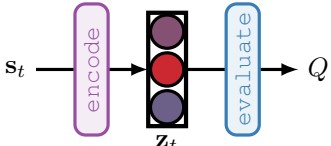

**Figure 1:** High-level structure of DQN.

In tree-search on-line planning, a look-ahead tree of possible future states is constructed by recursively applying an environment model. These states are typically evaluated by a heuristic, a learned value function, or Monte-Carlo rollouts. Backups through the tree aggregate these values along with the immediate rewards accumulated along each path to estimate the value of taking an action in the current state. This paper focuses on a simple tree-search with a deterministic transition function and no value uncertainty estimates, but our approach can be extended to tree-search variants like UCT (Kocsis & Szepesvári, 2006; Silver et al., 2016) if the components remain differentiable.

## 3 TREEQN

In this section, we propose TreeQN, a novel end-to-end differentiable tree-structured architecture for deep reinforcement learning. We first give an overview of the architecture, followed by details of each model component and the training procedure.

TreeQN uses a recursive tree-structured neural network between the encoded state $\mathbf{z}_t$ and the predicted state-action values $Q(\mathbf{s}_t, a)$, instead of directly estimating the state-action value from the current encoded state $\mathbf{z}_t$ using fully connected layers as in DQN (Mnih et al., 2015). Specifically, TreeQN uses a recursive model to refine its estimate of $Q(\mathbf{s}_t, a)$ via learned transition, reward, and value functions, and a tree backup (see Fig. 2). Because these learned components are shared throughout the tree, TreeQN implements an inductive bias, missing from DQN, that reflects the prior knowledge that the $Q$-values are properties of a stationary Markov process. We also encode the inductive bias that $Q$-values may be expressed as a sum of scalar rewards and values.

Specifically, TreeQN learns an action-dependent transition function that, given a state representation $\mathbf{z}_{l|t}$, predicts the next state representation $\mathbf{z}_{l+1|t}^{a_i}$ for action $a_i \in \mathcal{A}$, and the corresponding reward $\hat{r}_{l|t}^{a_i}$. To make the distinction between internal planning steps and steps taken in the environment explicit, we write $\mathbf{z}_{l|t}$ to denote the encoded state at time $t$ after $l$ internal transitions, starting with $\mathbf{z}_{0|t}$ for the encoding of $\mathbf{s}_t$. TreeQN applies this transition function recursively to construct a tree containing the state representations and rewards received for all possible sequences of actions up to some predefined depth $d$ ("Tree Transitioning" in Fig. 2).

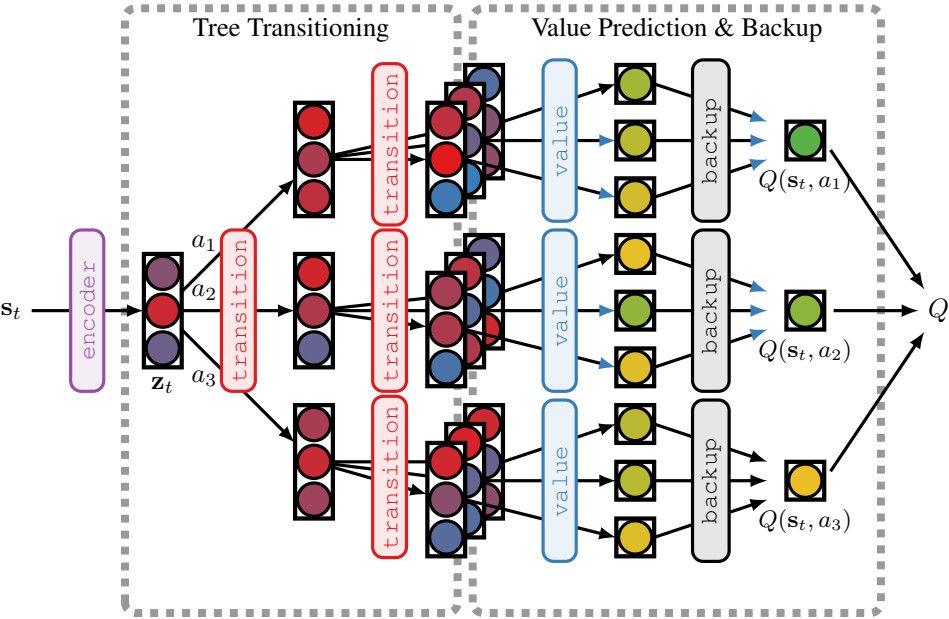

**Figure 2:** High-level structure of TreeQN with a tree depth of two and shared transition and evaluation functions (reward prediction and value mixing omitted for simplicity).

The value of each predicted state $V(\mathbf{z})$ is estimated with a value function module. Using these values and the predicted rewards, TreeQN then performs a tree backup, mixing the $k$-step returns along each path in the tree using TD($\lambda$) (Sutton, 1988; Sutton & Barto, 1998). This corresponds to "Value Prediction & Backup" in Fig. 2 and can be formalized as

$$Q^l(\mathbf{z}_{l|t}, a_i) = r(\mathbf{z}_{l|t}, a_i) + \gamma V^{(\lambda)}(\mathbf{z}_{l+1|t}) \tag{3}$$

$$V^{(\lambda)}(\mathbf{z}_{l|t}) = \begin{cases} V(\mathbf{z}_{l|t}^{a_i}) & l = d \\ (1 - \lambda)V(\mathbf{z}_{l|t}^{a_i}) + \lambda\, \mathtt{b}(Q^{l+1}(\mathbf{z}_{l+1|t}^{a_i}, a_j)) & l < d \end{cases} \tag{4}$$

where $\mathtt{b}$ is a function to recursively perform the backup. For $0 < \lambda < 1$, value estimates of the intermediate states are mixed into the final $Q$-estimate, which encourages the intermediate nodes of the tree to correspond to meaningful states, and reduces the impact of outlier values.

When $\lambda = 1$, and $\mathtt{b}$ is the standard hard $\mathtt{max}$ function, then Eq. 3 simplifies to a backup through the tree using the familiar Bellman equation:

$$Q(\mathbf{z}_{l|t}, a_i) = r(\mathbf{z}_{l|t}, a_i) + \begin{cases} \gamma V(\mathbf{z}_{d|t}^{a_i}) & l = d - 1 \\ \gamma \max_{a_j} Q(\mathbf{z}_{l+1|t}^{a_i}, a_j) & l < d - 1. \end{cases} \tag{5}$$

We note that even for a tree depth of only one, TreeQN imposes a significant structure on the value function by decomposing it as a sum of action-conditional reward and next-state value, and using a shared value function to evaluate each next-state representation.

Crucially, during training we backpropagate all the way from the final $Q$-estimate, through the value prediction, tree transitioning, and encoding layers of the tree, *i.e.*, the entire network shown in Fig. 2. Learning these components jointly ensures that they are useful for planning on-line.

### 3.1 MODEL COMPONENTS

In this section, we describe each of TreeQN's components in more detail.

**Encoder function.** As in DQN, a series of convolutional layers produces an embedding of the observed state, $\mathbf{z}_{0|t} = \mathtt{encode}(\mathbf{s}_t)$.

**Transition function.** We first apply a single fully connected layer to the current state embedding, shared by all actions. This generates an intermediate representation ($\mathbf{z}_{l+1|t}^{\text{env}}$) that could carry information about action-agnostic changes to the environment. In addition, we use a fully connected layer per action, which is applied to the intermediate representation to calculate a next-state representation that carries information about the effect of taking action $a_i$. We use residual connections for these layers:

$$\mathbf{z}_{l+1|t}^{\text{env}} = \mathbf{z}_{l|t} + \tanh(\boldsymbol{W}^{\text{env}}\mathbf{z}_{l|t} + \mathbf{b}^{\text{env}}),$$
$$\mathbf{z}_{l+1|t}^{a_i} = \mathbf{z}_{l+1|t}^{\text{env}} + \tanh(\boldsymbol{W}^{a_i}\mathbf{z}_{l+1|t}^{\text{env}}), \tag{6}$$

where $\boldsymbol{W}^{a_i}, \boldsymbol{W}^{\text{env}} \in \mathbb{R}^{k \times k}, \mathbf{b}^{\text{env}} \in \mathbb{R}^k$ are learnable parameters. Note that the next-state representation is calculated for every action $a_i$ independently using the respective transition matrix $\boldsymbol{W}^{a_i}$, but this transition function is shared for the same action throughout the tree.

A caveat is that the model can still learn to use different parts of the latent state space in different parts of the tree, which could undermine the intended parameter sharing in the model structure. To help TreeQN learn useful transition functions that maintain quality and diversity in their latent states, we introduce a unit-length projection of the state representations by simply dividing a state's vector representation by its L2 norm before each application of the transition function, $\mathbf{z}_{l|t} := \mathbf{z}_{l|t}/\left\|\mathbf{z}_{l|t}\right\|$. This prevents the magnitude of the representation from growing or shrinking, which encourages the behaviour of the transition function to be more consistent throughout the tree.

**Reward function.** In addition to predicting the next state, we also predict the immediate reward for every action $a_i \in \mathcal{A}$ in state $\mathbf{z}_{l|t}$ using

$$\hat{\mathbf{r}}(\mathbf{z}_{l|t}) = \boldsymbol{W}_2^r \texttt{ReLU}(\boldsymbol{W}_1^r \mathbf{z}_{l|t} + \mathbf{b}_1^r) + \mathbf{b}_2^r, \tag{7}$$

where $\boldsymbol{W}_1^r \in \mathbb{R}^{m \times k}, \boldsymbol{W}_2^r \in \mathbb{R}^{|\mathcal{A}| \times m}$ and $\texttt{ReLU}$ is the rectified linear unit (Nair & Hinton, 2010), and the predicted reward for a particular action $\hat{r}_{l|t}^{a_i}$ is the $i$-th element of the vector $\hat{\mathbf{r}}(\mathbf{z}_{l|t})$.

**Value function.** The value of a state representation $\mathbf{z}$ is estimated as

$$V(\mathbf{z}) = \mathbf{w}^\top\mathbf{z} + b, \tag{8}$$

where $\mathbf{w} \in \mathbb{R}^k$.

**Backup function.** We use the following function that can be recursively applied to calculate the tree backup:

$$\mathtt{b}(\mathbf{x}) = \sum_i x_i \operatorname{softmax}(\mathbf{x})_i. \tag{9}$$

Using a hard $\max$ for calculating the backup would result in gradient information only being used to update parameters along the maximal path in the tree. By contrast, the $\operatorname{softmax}$ allows us to use downstream gradient information to update parameters along all paths. Furthermore, it potentially reduces the impact of outlier value predictions. With a learned temperature for the $\operatorname{softmax}$, this function could represent the hard $\max$ arbitrarily closely. However, we did not find an empirical difference so we left the temperature at $1$.

## 3.2 GROUNDING THE MODEL COMPONENTS

The TreeQN architecture is fully differentiable, so we can directly use it in the place of a $Q$-function in any deep RL algorithm with discrete actions. Differentiating through the entire tree ensures that the learned components are useful for planning on-line, as long as that planning is performed in the same way as during training.

However, it seems plausible that auxiliary objectives based on minimising the error in predicting rewards or observations could improve the performance by helping to ground the transition and reward functions to the environment. It could also encourage TreeQN to perform model-based planning in an interpretable manner. In principle, such objectives could give rise to a spectrum of methods from model-free to fully model-based. At one extreme, TreeQN without auxiliary objectives can be seen as a model-free approach that draws inspiration from tree-search planning to encode valuable inductive biases into the neural network architecture. At the other extreme, perfect, grounded reward and transition models could in principle be learned. Using them in our architecture would then correspond

to standard model-based lookahead planning. The sweet spot could be an intermediate level of grounding that maintains the flexibility of end-to-end model-free learning while benefiting from the additional supervision of explicit model learning. To investigate this spectrum, we experiment with two auxiliary objectives.

**Reward grounding.** We experiment with an L2 loss regressing $\hat{r}_{l|t}^{a_{t:t+l-1}}$, the predicted reward at level $l$ of the tree corresponding to the selected action sequence $\{a_t \dots a_{t+l-1}\}$, to the true observed rewards. For each of the $n$ timesteps of $n$-step Q-learning this gives:

$$\mathcal{L} = \mathcal{L}_{\text{nstep-Q}} + \eta_r \sum_{\text{envs}} \sum_{j=1}^{n} \sum_{l=1}^{\bar{d}} \left( \hat{r}_{l|t+j}^{a_{t+j:t+j+l-1}} - r_{t+j+l-1} \right)^2, \tag{10}$$

where $\eta_r$ is a hyperparameter weighting the loss, and $\bar{d} = \min(d, n - j + 1)$ restricts the sum to rewards for which we have already observed the true value.

**State grounding.** We experiment with a grounding in the latent space, using an L2 loss to regress the predicted latent state $\mathbf{z}_{l|t}^{a_{t:t+l}}$ at level $l$ of the tree to $\mathbf{z}_{0|t+l}$, the initial encoding of the true state corresponding to the actions actually taken:

$$\mathcal{L} = \mathcal{L}_{\text{nstep-Q}} + \eta_s \sum_{\text{envs}} \sum_{j=1}^{n} \sum_{l=1}^{\bar{d}} \left( \mathbf{z}_{l|t+j}^{a_{t+j:t+j+l-1}} - \mathbf{z}_{0|t+j+l} \right)^2. \tag{11}$$

By employing an additional decoder module, we could use a similar loss to regress decoded observations to the true observations. In informal experiments, joint training with such a decoder loss did not yield good performance, as also found by Oh et al. (2017).

In Section 7.1, we present results on the use these objectives, showing that reward grounding gives better performance, but that our method for state grounding does not.

## 4 ATREEC

The intuitions guiding the design of TreeQN are as applicable to policy search as to value-based RL, in that a policy can use a tree planner to improve its estimates of the optimal action probabilities (Gelly & Silver, 2007; Silver et al., 2017a). As our proposed architecture is trained end-to-end, it can be easily adapted for use as a policy network.

In particular, we propose ATreeC, an actor-critic extension of TreeQN. In this architecture, the policy network is identical to TreeQN, with an additional softmax layer that converts the $Q$ estimates into the probabilities of a stochastic policy.

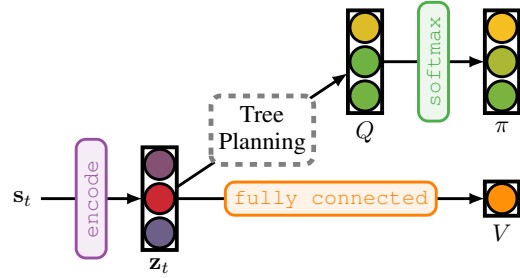

**Figure 3:** High-level structure of ATreeC.

The critic shares the encoder parameters, and predicts a scalar state value with a single fully connected layer: $V_{\text{cr}}(\mathbf{s}) = \mathbf{w}_{\text{cr}}^\top \mathbf{z} + b_{\text{cr}}$. We used different parameters for the critic value function and the actor's tree-value-function module, but found that sharing these parameters had little effect on performance. The entire setup, shown in Fig. 3, is trained with A2C as described in Section 2, with the addition of the same auxiliary losses used for TreeQN. Note that TreeQN could also be used in the critic, but we leave this possibility to future work.

## 5 RELATED WORK

There is a long history of work combining model-based and model-free RL. An early example is Dyna-Q (Sutton, 1990) which trains a model-free algorithm with samples drawn from a learned model. Similarly, van Seijen et al. (2011) train a sparse model with some environment samples that can be used to refine a model-free $Q$-function. Gu et al. (2016) use local linear models to generate

additional samples for their model-free algorithm. However, these approaches do not attempt to use the model on-line to improve value estimates.

In deep RL, *value iteration networks* (Tamar et al., 2016) use a learned differentiable model to plan on the fly, but require planning over the full state space, which must also possess a spatial structure with local dynamics such that convolution operations can execute the planning algorithm.

The *predictron* (Silver et al., 2017b) instead learns abstract-state transition functions in order to predict values. However, it is restricted to policy evaluation without control. *Value prediction networks* (VPNs, Oh et al., 2017) take a similar approach but are more closely related to our work because the learned model components are used in a tree for planning. However, in their work this tree is only used to construct targets and choose actions, and not to compute the value estimates during training. Such estimates are instead produced from non-branching trajectories following on-policy action sequences. By contrast, TreeQN is a unified architecture that constructs the tree dynamically at every timestep and differentiates through it, eliminating any mismatch between the model at training and test time. Furthermore, we do not use convolutional transition functions, and hence do not impose spatial structure on the latent state representations. These differences simplify training, allow our model to be used more flexibly in other training regimes, and explain in part our substantially improved performance on the Atari benchmark.

Donti et al. (2017) propose differentiating through a stochastic programming optimisation using a probabilistic model to learn model parameters with respect to their true objective rather than a maximum likelihood surrogate. However, they do not tackle the full RL setting, and do not use the model to repeatedly or recursively refine predictions.

*Imagination-augmented agents* (Weber et al., 2017) learn to improve policies by aggregating rollouts predicted by a model. However, they rely on pretraining an observation-space model, which we argue will scale poorly to more complex environments. Further, their aggregation of rollout trajectories takes the form of a generic RNN rather than a value function and tree backup, so the inductive bias based on the structure of the MDP is not explicitly present.

A class of *value gradient* methods (Deisenroth & Rasmussen, 2011; Fairbank & Alonso, 2012; Heess et al., 2015) also differentiates through models to train a policy. However, this approach does not use the model during execution to refine the policy, and requires continuous action spaces.

Oh et al. (2015) and Chiappa et al. (2017) propose methods for learning observation-prediction models in the Atari domain, but use these models only to improve exploration. Variants of scheduled sampling (Bengio et al., 2015) may be used to improve robustness of these models, but scaling to complex domains has proven challenging (Talvitie, 2014).

## 6  EXPERIMENTS

We evaluate TreeQN and ATreeC in a simple box-pushing environment, as well as on the subset of nine Atari environments that Oh et al. (2017) use to evaluate VPN. The experiments are designed to determine whether or not TreeQN and ATreeC outperform DQN, A2C, and VPN, and whether they can scale to complex domains. We also investigate how to best ground the the transition function with auxiliary losses. Furthermore, we compare against alternative ways to increase the number of parameters and computations of a standard DQN architecture, and study the impact of tree depth. Full details of the experimental setup, as well as architecture and training hyperparameters, are given in the appendix.

**Grounding.** We perform a hyperparameter search over the coefficients $\eta_r$ and $\eta_s$ of the reward and state grounding auxiliary losses, on the Atari environment Seaquest. These experiments aim to determine the relevant trade-offs between the flexibility of a model-free approach and the potential benefits of a more model-based algorithm.

**Box Pushing.** We randomly place an agent, 12 boxes, 5 goals and 6 obstacles on the center $6 \times 6$ tiles of an $8 \times 8$ grid. The agent's goal is to push boxes into goals in as few steps as possible while avoiding obstacles. Boxes may not be pushed into each other. The obstacles, however, are 'soft' in that they are do not block movement, but generate a negative reward if the agent or a box moves onto an obstacle. This rewards better planning without causing excessive gridlock. This environment is inspired by Sokoban, as used by Weber et al. (2017), in that poor actions can generate irreversibly

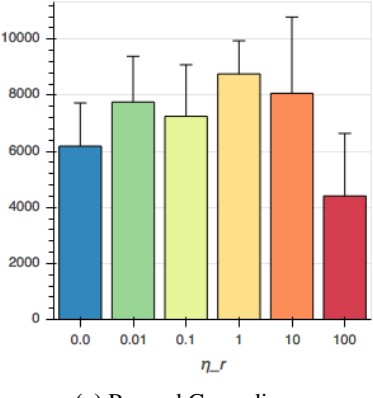 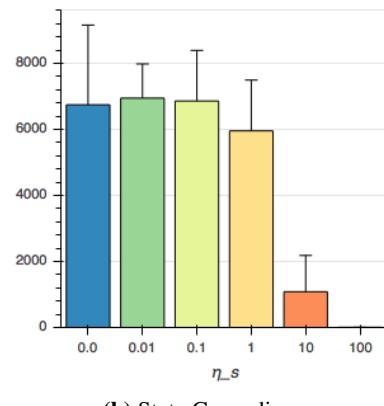

**(a)** Reward Grounding.        **(b)** State Grounding.

**Figure 4:** Grounding the reward and transition functions using auxiliary losses: final returns on Seaquest plotted against the coefficient of the auxiliary loss.

bad configurations. However, the level generation process for Sokoban is challenging to reproduce exactly and has not been open-sourced. More details of the environment and rewards are given in Appendix A.1.

**Atari.** To demonstrate the general applicability of TreeQN and ATreeC to complex environments, we evaluate them on the Atari 2600 suite (Bellemare et al., 2013). Following Oh et al. (2017), we use their set of nine environments and a frameskip of 10 to facilitate planning over reasonable timescales.

TreeQN adds additional parameters to a standard DQN architecture. We compare TreeQN to two baseline architectures with increased computation and numbers of parameters to verify the benefit of the additional structure and grounding. *DQN-Wide* doubles the size of the embedding dimension (1024 instead of 512). *DQN-Deep* inserts two additional fully connected layers with shared parameters and residual connections between the two fully-connected layers of DQN. This is in effect a non-branching version of the TreeQN architecture that also lacks explicit reward prediction.

## 7 RESULTS & DISCUSSION

In this section, we present our experimental results for TreeQN and ATreeC.

### 7.1 GROUNDING

Fig. 4 shows the result of a hyperparameter search on $\eta_r$ and $\eta_s$, the coefficients of the auxiliary losses on the predicted rewards and latent states. An intermediate value of $\eta_r$ helps performance but there is no benefit to using the latent space loss. Subsequent experiments use $\eta_r = 1$ and $\eta_s = 0$.

The predicted rewards that the reward-grounding objective encourages the model to learn appear both in its own $Q$-value prediction and in the target for $n$-step $Q$-learning. Consequently, we expect this auxiliary loss to be well aligned with the true objective. By contrast, the state-grounding loss (and other potential auxiliary losses) might help representation learning but would not explicitly learn any part of the desired target. It is possible that this mismatch between the auxiliary and primary objective leads to degraded performance when using this form of state grounding. One potential route to overcoming this obstacle to joint training would be pre-training a model, as done by Weber et al. (2017). Inside TreeQN this model could then be fine-tuned to perform well inside the planner. We leave this possiblity to future work.

### 7.2 BOX PUSHING

Fig. 5a shows the results of TreeQN with tree depths 1, 2, and 3, compared to a DQN baseline. In this domain, there is a clear advantage for the TreeQN architecture over DQN. TreeQN learns policies that are substantially better at avoiding obstacles and lining boxes up with goals so they can be easily

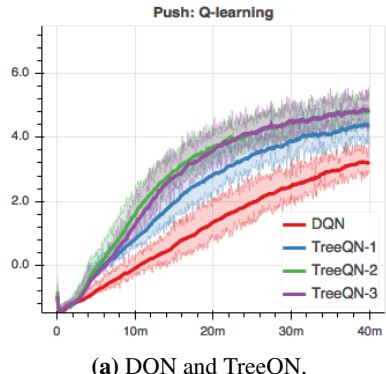
**(a)** DQN and TreeQN.

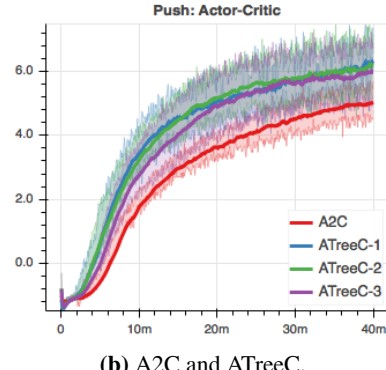
**(b)** A2C and ATreeC.

**Figure 5:** Box-pushing results: the $x$-axis shows the number of transitions observed across all of the synchronous environment threads.

pushed in later. TreeQN also substantially speeds up learning. We believe that the greater structure brought by our architecture regularises the model, encouraging appropriate state representations to be learned quickly. Even a depth-1 tree improves performance significantly, as disentangling the estimation of rewards and next-state values makes them easier to learn. This is further facilitated by the sharing of value-function parameters across branches.

When trained with $n$-step Q-learning, the deeper depth-2 and depth-3 trees learn faster and plateau higher than the shallow depth-1 tree. In the this domain, useful transition functions are relatively easy to learn, and the extra computation time with those transition modules can help refine value estimates, yielding advantages for additional depth.

Fig. 5b shows the results of ATreeC with tree depths 1, 2, and 3, compared to an A2C baseline. As with TreeQN, ATreeC substantially outperforms the baseline. Furthermore, thanks to its stochastic policy, it substantially outperforms TreeQN. Whereas TreeQN and DQN sometimes indecisively bounce back and forth between adjacent states, ATreeC captures this uncertainty in its policy probabilities and thus acts more decisively. However, unlike TreeQN, ATreeC shows no pronounced differences for different tree depths. This is in part due to a ceiling effect in this domain. However, ATreeC is also gated by the quality of the critic's value function, which in these experiments was a single linear layer after the state encoding as described in Section 4. Nonetheless, this result demonstrates the ease with which TreeQN can be used as a drop-in replacement for any deep RL algorithm that learns policies or value functions for discrete actions.

## 7.3 ATARI

Table 1 summarises all our Atari results, while Fig. 6 shows learning curves in depth. TreeQN shows substantial benefits in many environments compared to our DQN baseline, which itself often outperforms VPN (Oh et al., 2017). ATreeC always matches or outperforms A2C. We present the mean performance of five random seeds, while the VPN results reported by Oh et al. (2017), shown as dashed lines in Fig. 6, are the mean of the *best* five seeds of an unspecified number of trials.

**TreeQN.** In all environments except Frostbite, TreeQN outperforms DQN on average, with the most significant gains in Alien, CrazyClimber, Enduro, Krull, and Seaquest. Many of these environments seem well suited to short horizon look-ahead planning, with simple dynamics that generalise well and tradeoffs between actions that become apparent only after several timesteps. For example, an incorrect action in Alien can trap the agent down a corridor with an alien. In Seaquest, looking ahead could help determine whether it is better to go deeper to collect more points or to surface for oxygen. However, even in a game with mostly reactive decisions like the racing game Enduro, TreeQN shows significant benefits.

TreeQN also outperforms the additional baselines of DQN-Wide and DQN-Deep, indicating that the additional structure and grounding of our architecture brings benefits beyond simply adding model capacity and computation. In particular, it is interesting that DQN-Deep is often outperformed by the vanilla DQN baseline, as optimisation difficulties grow with depth. In contrast, the additional

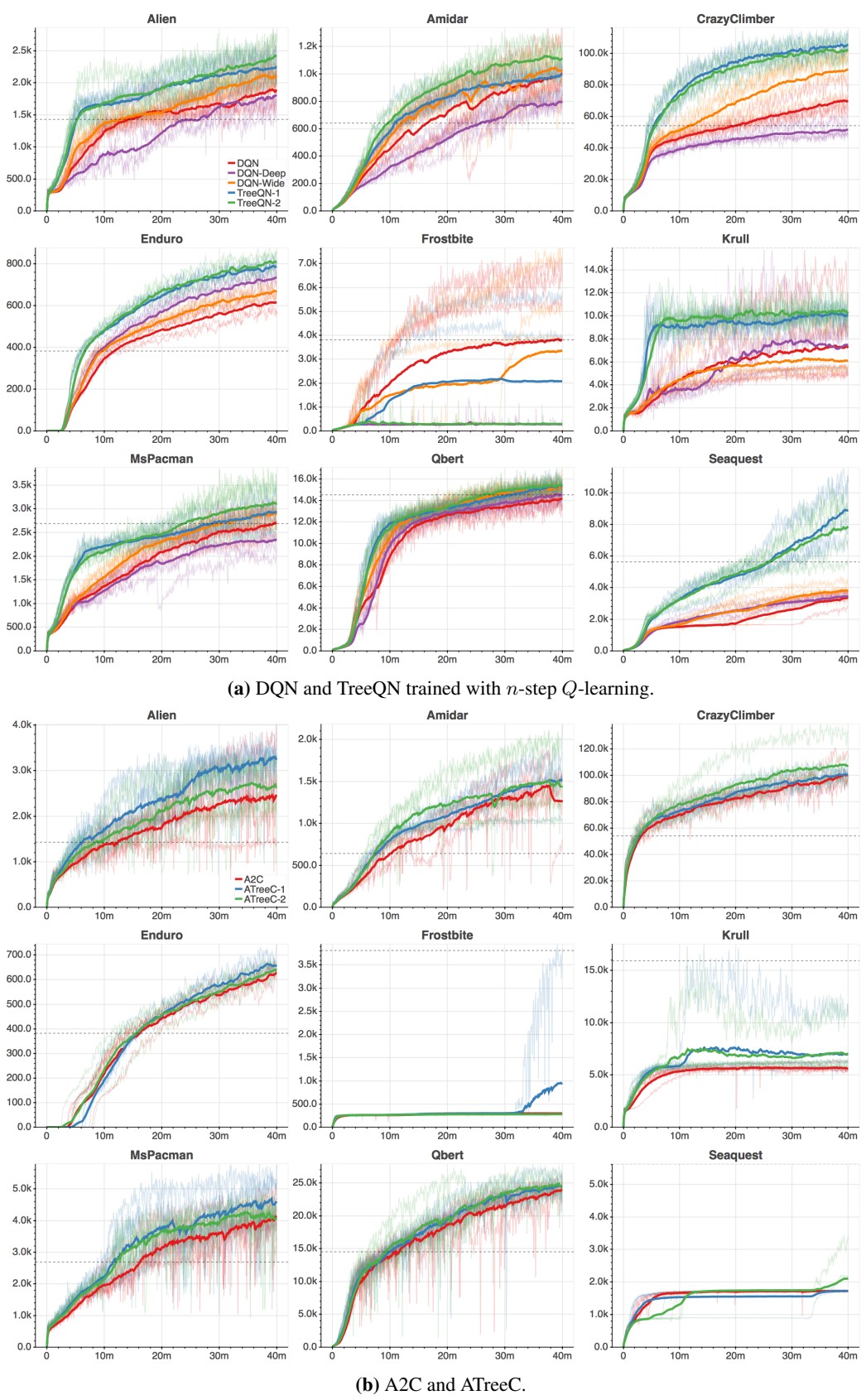

**(a)** DQN and TreeQN trained with $n$-step $Q$-learning.

**(b)** A2C and ATreeC.

**Figure 6:** Results for the Atari domain. The y-axis shows the moving average over 100 episodes. Each of five random seeds is plotted faintly, with the mean in bold.

| | Alien | Amidar | Crazy Climber | Enduro | Frostbite | Krull | Ms. Pacman | Q*Bert | Seaquest |
|---|---|---|---|---|---|---|---|---|---|
| DQN (Oh et al., 2017) | 1804 | 535 | 41658 | 326 | 3058 | 12438 | 2804 | 12592 | 2951 |
| VPN (Oh et al., 2017) | 1429 | 641 | 54119 | 382 | 3811 | **15930** | 2689 | 14517 | 5628 |
| $n$-step DQN | 1969 | 1033 | 71623 | 625 | **3968** | 7860 | 2774 | 14468 | 3465 |
| DQN-Deep | 1906 | 825 | 53101 | 745 | 493 | 8605 | 2410 | 15094 | 3575 |
| DQN-Wide | 2187 | 1074 | 91380 | 682 | 3493 | 6603 | 3061 | 15794 | 3909 |
| TreeQN-1 | 2321 | 1030 | 107983 | 800 | 2254 | 10836 | 3030 | 15688 | **9302** |
| TreeQN-2 | 2497 | 1170 | 104932 | **825** | 581 | 11035 | 3277 | 15970 | 8241 |
| A2C | 2673 | 1525 | 102776 | 642 | 297 | 5784 | 4352 | 24451 | 1734 |
| ATreeC-1 | **3448** | **1578** | 102546 | 678 | 1035 | 8227 | **4866** | 25159 | 1734 |
| ATreeC-2 | 2813 | 1566 | **110712** | 649 | 281 | 8134 | 4450 | **25459** | 2176 |

**Table 1:** Summary of Atari results. Each number is the best score throughout training, calculated as the mean of the last 100 episode rewards averaged over exactly five agents trained with different random seeds. Note that Oh et al. (2017) report the same statistic, but average instead over the *best* five of an unspecified number of agents.

structure and auxiliary loss employed by TreeQN turn its additional depth from a liability into a strength.

**ATreeC.** ATreeC matches or outperforms its baseline (A2C) in all environments. Compared to TreeQN, ATreeC's performance is better across most environments, particularly on Qbert, reflecting an overall advantage for actor-critic also found by Mnih et al. (2016) and in our box-pushing experiments. However, performance is much worse on Seaquest, revealing a deficiency in exploration as policy entropy collapses too rapidly and consequently the propensity of policy gradient methods to become trapped in a local optimum.

In Krull and Frostbite, most algorithms have poor performance, or high variance in returns from run to run, as agents are gated by their ability to explore. Both of these games require the completion of sub-levels in order to accumulate large scores, and none of our agents reliably explore beyond the initial stages of the game. Mean performance appears to favor TreeQN and ATreeC in Krull, and perhaps DQN in Frostbite, but the returns are too variable to draw conclusions from this number of random seeds. Combining TreeQN and ATreeC with smart exploration mechanisms is an interesting direction for future work to improve robustness of training in these types of environments.

Compared to the box-pushing domain, there is less of a clear performance difference between trees of different depths. In some environments (Amidar, MsPacman), greater depth does appear to be employed usefully by TreeQN to a small extent, resulting in the best-performing individual agents. However, for the Atari domain the embedding size for the transition function we use is much larger (512 compared to 128), and the dynamics are much more complex. Consequently, we expect that optimisation difficulties, and the challenge of learning abstract-state transition functions, impede the utility of deeper trees in some cases. We look to future work to further refine methods for learning to plan abstractly in complex domains. However, the decomposition of Q-value into reward and next-state value employed by the first tree expansion is clearly of utility in a broad range of tasks.

When inspecting the learned policies and trees, we find that the values sometimes correspond to intuitive reasoning about sensible policies, scoring superior action sequences above poorer ones. However, we find that the actions corresponding to branches of the tree that are scored most highly are frequently not taken in future timesteps. The flexibility of TreeQN and ATreeC allows our agents to find any useful way to exploit the computation in the tree to refine action-value estimates. As we found no effective way to strongly ground the model components without sacrificing performance, the interpretability of learned trees is limited.

# 8 CONCLUSIONS & FUTURE WORK

We presented TreeQN and ATreeC, new architectures for deep reinforcement learning in discrete-action domains that integrate differentiable on-line tree planning into the action-value function or policy. Experiments on a box-pushing domain and a set of Atari games show the benefit of these architectures over their counterparts, as well as over VPN. In future work, we intend to investigate enabling more efficient optimisation of deeper trees, encouraging the transition functions to produce interpretable plans, and integrating smart exploration.

ACKNOWLEDGMENTS

We thank Sasha Salter, Luisa Zintgraf, and Wendelin Böhmer for their contributions and valuable comments on drafts of this paper. This work was supported by the UK EPSRC CDT in Autonomous Intelligent Machines and Systems. This project has received funding from the European Research Council (ERC) under the European Union's Horizon 2020 research and innovation programme (grant agreement #637713). The NVIDIA DGX-1 used for this research was donated by the NVIDIA Corporation.

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

# A    APPENDIX

## A.1    BOX PUSHING

**Environment.** For each episode, a new level is generated by placing an agent, 12 boxes, 5 goals and 6 obstacles in the center $6 \times 6$ tiles of an $8 \times 8$ grid, sampling locations uniformly. The outer tiles are left empty to prevent initial situations where boxes cannot be recovered.

The agent may move in the four cardinal directions. If the agent steps off the grid, the episode ends and the agent receives a penalty of $-1$. If the agent moves into a box, it is pushed in the direction of movement. Moving a box out of the grid generates a penalty of $-0.1$. Moving a box into another box is not allowed and trying to do so generates a penalty of $-0.1$ while leaving all positions unchanged. When a box is pushed into a goal, it is removed and the agent receives a reward of $+1$.

Obstacles generate a penalty of $-0.2$ when the agent or a box is moved onto them. Moving the agent over goals incurs no penalty. Lastly, at each timestep the agent receives a penalty of $-0.01$. Episodes terminate when 75 timesteps have elapsed, the agent has left the grid, or no boxes remain.

The observation is given to the model as a tensor of size $5 \times 8 \times 8$. The first four channels are binary encodings of the position of the agent, goals, boxes, and obstacles respectively. The final channel is filled with the number of timesteps remaining (normalised by the total number of timesteps allowed).

**Architecture.** The encoder consists of (conv-3x3-1-24, conv-3x3-1-24, conv-4x4-1-48, fc-128), where conv-wxh-s-n denotes a convolution with $n$ filters of size $w \times h$ and stride $s$, and fc-h denotes a fully connected layer with $h$ hidden units. All layers are separated with ReLU nonlinearities. The hidden layer of the reward function MLP has 64 hidden units.

## A.2    ATARI

Preprocessing of inputs follows the procedure of Mnih et al. (2015), including concatenation of the last four frames as input, although we use a frameskip of 10.

**Architecture.** The Atari experiments have the same architecture as for box-pushing, except for the encoder architecture which is as follows: (conv-8x8-4-16, conv-4x4-2-32, fc-512).

## A.3    OTHER HYPERPARAMETERS

All experiments use RMSProp (Tieleman & Hinton, 2012) with a learning rate of 1e-4, a decay of $\alpha = 0.99$, and $\epsilon = $ 1e-5.

The learning rate was tuned coarsely by running DQN on the Seaquest environment, and kept the same for all subsequent experiments (box-pushing and Atari).

For DQN and TreeQN, $\epsilon$ for $\epsilon$-greedy exploration was decayed linearly from 1 to 0.05 over the first 4 million environment transitions observed (after frameskipping, so over 40 million atomic Atari timesteps).

For A2C and ATreeC, we use a value-function loss coefficient $\alpha = 0.5$ and an entropy regularisation $\beta = 0.01$.

The reward prediction loss was scaled by $\eta_r = 1$.

We use $n_{\text{steps}} = 5$ and $n_{\text{envs}} = 16$, for a total batch size of 80.

The discount factor is $\gamma = 0.99$ and the target networks are updated every $40,000$ environment transitions.

