# OpenReview forum: "TreeQN and ATreeC: Differentiable Tree-Structured Models for Deep Reinforcement Learning"
_ICLR.cc/2018/Conference — Accept (Poster)_

### Official Review · AnonReviewer1 · 2017-11-27
**Interesting proposal, well-written, some short-comings in the results.**

**Rating:** 8
**Confidence:** 5

**Review:**

The authors propose a new network architecture for RL that contains some relevant inductive biases about planning. This fits into the recent line of work on implicit planning where forms of models are learned to be useful for a prediction/planning task. The proposed architecture performs something analogous to a full-width tree search using an abstract model (learned end-to-end). This is done by expanding all possible transitions to a fixed depth before performing a max backup on all expanded nodes. The final backup value is the Q-value prediction for a given state, or can represent a policy through a softmax.

I thought the paper was clear and well-motivated. The architecture (and various associated tricks like state vector normalization) are well-described for reproducibility.

Experimental results seem promising but I wasn’t fully convinced of its conclusions. In both domains, TreeQN and AtreeC are compared to a DQN architecture, but it wasn’t clear to me that this is the right baseline. Indeed TreeQN and AtreeC share the same conv stack in the encoder (I think?), but also have the extra capacity of the tree on top. Can the performance gain we see in the Push task as a function of tree depth be explained by the added network capacity? Same comment in Atari, but there it’s not really obvious that the proposed architecture is helping. Baselines could include unsharing the weights in the tree, removing the max backup, having a regular MLP with similar capacity, etc.

Page 5, the auxiliary loss on reward prediction seems appropriate, but it’s not clear from the text and experiments whether it actually was necessary. Is it that makes interpretability of the model easier (like we see in Fig 5c)? Or does it actually lead to better performance?

Despite some shortcomings in the result section, I believe this is good work and worth communicating as is.

---

> ### Author Response · Authors · 2017-12-12
> **TreeQN/ATreeC incorporate a valuable inductive bias and simply adding more width or depth to DQN does not necessarily improve performance**
>
> Thank you for your positive comments and useful feedback.
>
> Concerning baselines, our preliminary experiments showed that simply adding more parameters via width or depth to a DQN architecture did not result in significant performance gains, which we will make clear in the paper. A large-scale investigation of such architectures and their combination with auxiliary losses on many Atari games may be infeasible for us, but we have added to the appendix a figure demonstrating the limitations of naively adding parameters to DQN on the box-pushing domain.
>
> We didn’t do a systematic investigation of the reward prediction loss across all environments, but in preliminary experiments on Seaquest and the box-pushing environment it helped performance. Interpretable sequences for box-pushing tended to appear when rewards were immediately available, which leads us to believe the grounding from this loss played a part.

---

### Official Review · AnonReviewer2 · 2017-11-27

**Rating:** 4
**Confidence:** 5

**Review:**

# Update after the rebuttal
Thank you for the rebuttal.
The authors claim that the source of objective mismatch comes from n-step Q-learning, and their method is well-justified in 1-step Q-learning. However, there is still a mismatch even with 1-step Q-learning because the bootstrapped target is also computed from the TreeQN. More specifically, there can be a mismatch between the optimal action sequences computed from TreeQN at time t and t+1 if the depth of TreeQN is equal or greater than 2. Thus, the author's response is still not convincing to me.
I like the overall idea of using a tree-structured neural network which internally performs planning as an abstraction of Q-function, which makes implementation simpler compared to VPN. However, the particular method (TreeQN) proposed in this paper introduces a mismatch in the model learning as mentioned above. One could argue that TreeQN is learning an "abstract" planning rather than "grounded" planning. However, the fact that reward prediction loss is used to train TreeQN significantly weakens this claim, and there is no such an evidence in the paper.
In conclusion, I think the research direction is worth pursuing, but the proposed modification from VPN is not well-justified.

# Summary
This paper proposes TreeQN and ATreeC which perform look-ahead planning using neural networks. TreeQN simulates the future by predicting rewards/values of the future states and performs tree backup to construct Q-values. ATreeC is an actor-critic architecture that uses a softmax over TreeQN. The architecture is trained through n-step Q-learning with reward prediction loss. The proposed methods outperform DQN baseline on 2D Box Pushing domain and outperforms VPN on Atari games.

[Pros]
- The paper is easy to follow.
- The application to actor-critic setting (ATreeC) is novel, though the underlying idea was proposed by [O'Donoghue et al., Schulman et al.].

[Cons]
- The proposed method has a technical issue.
- The proposed idea (TreeQN) and underlying motivation are almost same as those of VPN [Oh et al.], but there is no in-depth discussion that shows why TreeQN is potentially better than VPN.
- Comparison to VPN on Atari is not much convincing.

# Novelty and Significance
- The underlying motivation (planning without predicting observations), the architecture (transition/reward/value functions applied to the latent state space), and the algorithm (n-step Q-learning with reward prediction loss) are same as those of VPN. But, the paper does not provide in-depth discussion on this. The following is the differences that I found from this paper, so it would be important to discuss why such differences are important.

1) The paper emphasizes the "fully-differentiable tree planning" aspect in contrast to VPN that back-propagates only through "non-branching" trajectories during training. However, differentiating TreeQN also amounts to back-propagating through a "single" trajectory in the tree that gives the maximum Q-value. Thus, the only difference between TreeQN and VPN is that TreeQN follows the best (estimated) action sequence, while VPN follows the chosen action sequence in retrospect during back-propagation. Can you justify why following the best estimated action sequence is better than following the chosen action sequence during back-propagation (see Technical Soundness section for discussion)?

2) TreeQN only sets targets for the final Q-value after tree backup, whereas VPN sets targets for all intermediate value predictions in the tree. Why is TreeQN's approach better than VPN's approach?

- The application to actor-critic setting (ATreeC) is novel, though the underlying idea of combining Q-learning with policy gradient was proposed by [O'Donoghue et al.] and [Schulman et al.].

# Technical Soundness
- The proposed idea of setting targets for the final Q-value after tree backup can potentially make the temporal credit assignment difficult, because the best estimated actions during tree planning does not necessarily match with the chosen actions. Suppose that TreeQN estimated "up-right-right" as the best future action sequence the during 3-step tree planning, while the agent actually ended up with choosing "up-left-left" (this is possible because the agent re-plans at every step and follows epsilon-greedy policy). Following n-step Q-learning procedure, we end up with setting target Q-value based on on-policy action sequence "up-left-left", while back-propagating through "up-right-right" action sequence in the TreeQN's plan. This causes a wrong temporal credit assignment, because TreeQN can potentially increase/decrease value estimates in the wrong direction due to the mismatch between the planned actions and chosen actions. So, it is unclear why the proposed algorithm is technically correct or better than VPN's approach (i.e., back-propagating through the chosen actions in the search tree).

# Quality
- Comparison to VPN on Atari is not convincing because TreeQN-1 is actually (almost) equivalent to VPN-1, but the results show that TreeQN-1 performs much better than VPN on many games. Since the authors took the numbers from [Oh et al.] rather than replicating VPN, it is possible that the gap comes from implementation details (e.g., hyperparameter).

# Clarity
- The paper is overall easy to follow and the description of the proposed method is clear.

---

> ### Author Response · Authors · 2017-12-12
> **TreeQN is more general than VPN -- it can be used as drop in replacement for *any* value estimation, enabling e.g. ATreeC, and matching train/test contexts**
>
> Thank you for your feedback.
>
> Regarding the soundness of n-step Q-learning targets:
> As you point out, there is a mismatch between the n-step targets, which include an on-policy component, and our model’s estimates of the optimal Q-function. However, this mismatch appears for *any* model estimating the optimal Q* with partially on-policy bootstraps. The weakly grounded internal temporal semantics of our architecture do not exacerbate this problem, but simply render more explicit the mechanism for estimating Q*.
> In 1-step Q-learning, or when using policy gradients, this model-objective mismatch does not appear. In practice, n-step Q-learning targets help to stabilise and speed learning despite the unusual combination of on- and off- policy components in the targets. However, it is true that 1-step Q-learning, or policy gradients, provides objectives more consistent with our overall approach.
>
> Regarding the comparison to VPN algorithmically:
> Following the best estimated action sequence removes a mismatch between the use of the model components at training and test-time: in our approach, the components are freely optimised for their use in estimating the optimal action-value or action-probability, rather than trained to match a greedy bootstrap with an on-policy internal path. We believe it is crucial to maintain an equivalent context at training and evaluation time.
> This is also the motivation for optimising Q only after the full tree backup, rather than also after partial backups. We want to learn a model that is as good as possible within the specific architecture we use, rather than across a class of related architectures with varying tree-depths. It is possible that such transfer learning could help in some problems. However, it is important to note that intermediate value estimates are still used, as they are mixed into the final prediction during the backup.
> Constructing the full tree at each time-step frees us to make value estimates for all (root-node) actions, which enables the extension to ATreeC -- something that can’t easily be done with VPN. This extension is more about using an *architecture* designed for value-prediction in the context of policy gradients, rather than using *algorithmic* components of Q-learning with policy gradients as in the work of [O'Donoghue et al., Schulman et al.].
> Our overall strategy also simplifies training, as the whole model can be used as a drop-in replacement for an action-value or policy network, without recombining the components in a different manner for on-policy training segments, target evaluation, and testing. In our view this is a valuable contribution over VPN.
>
> Regarding the experimental comparison to VPN, it is clear that some details of hyperparameters or implementation constitute a large part of the difference (as most clearly seen in the different baseline DQN results). This is precisely why we focus on comparing to our own, much stronger, DQN and A2C baselines. We included these data to facilitate other work using frameskip-10 Atari as a domain for planning-inspired deep RL. We feel it is unreasonable to expect a reimplementation of VPN with tuning to approach the level of our baselines, and assume the authors of that work put a reasonable effort into optimising their algorithm.

---

### Official Review · AnonReviewer3 · 2017-11-29

**Rating:** 5
**Confidence:** 3

**Review:**


 This was an interesting read.

I feel that there is a mismatch between intuition of what a model could do (based on the structure of the architecture) versus what a model does. Just because the transition function is shared and the model could learn to construct a tree, when trained end-to-end the system is not sufficiently constrained to learn this specific behaviour. More to a point. I think the search tree perspective is interesting, but isn’t this just a deeper model with shared weights? And a max operation? It seems no loss is used to force the embeddings produced by the transition model to match the embeddings that you would get if you take a particular action in a particular state, right? Is there any specific attempt to visualize or understand the embeddings inside the tree? The same regarding the rewards. If there is no auxiliary loss attempting to force the intermediary prediction to be valid rewards, why would the model use those free latent variables to encode rewards? I think this is a pitfall that many deep network papers fall, where by laying out a particular structure it is directly inferred that the model discovers or follows a particular solution (where the latent have prescribed semantics). I would argue that is rarely the case. When the system is learned end-to-end, the structure does not impose the behaviour of the model, and is up to the authors of the paper to prove that the trained model does anything similar to expanding a tree. And this is not by showing final performance on a game. If indeed the model does anything similar to search, than all intermediary representations should correspond to what semantically they should.
Ignoring my verbose comment, another view is that the baseline are disadvantaged to the treeQN, because they have less parameters (and are less deep which has a huge impact on the learnability and expressivity of the deep network).

---

> ### Author Response · Authors · 2017-12-12
> **Our model strikes a balance between grounding the transition function and performance in complex environments**
>
> It’s great to hear that you found this work interesting - and thank you for your feedback.
>
> The goal of our research is to design methods that yield good performance in the considered tasks, and we hope the reviewers will evaluate our paper accordingly. Explicit grounding, or lack thereof, is merely a means to the end of maximising task performance. Therefore, while the empirical question of how explicitly to ground the model is fascinating, we believe the quality of the paper should not be measured in terms of how explicitly the model is grounded. See also the answer to the anonymous comment from 21st of November below.
>
> We found that grounding the reward function (which takes intermediate embeddings as input) with an explicit loss did help performance, so we included that in the objective (see Sec. 3.2). However, we found that explicitly grounding the latent representations based on a reconstruction loss did not help. In the paper, we also discuss reasons why one should avoid such objectives when constructed in the observation space. Also note that intermediate value predictions are mixed into the final prediction. While this doesn’t force a grounding, it encourages each of the intermediate embeddings to correspond to a valid state embedding.
>
> The key idea behind our paper is that the architecture that makes sense for a grounded model (e.g., tree-planning) should still provide a useful inductive bias for a learned model that is only weakly grounded or not grounded at all.
>
> Concerning baselines, our preliminary experiments showed that simply adding more parameters via width or depth to a DQN architecture did not result in significant performance gains, which we will make clear in the paper. A large-scale investigation of such architectures and their combination with auxiliary losses on many Atari games may be infeasible for us, but we have added to the appendix a figure demonstrating the limitations of naively adding parameters to DQN on the box-pushing domain.

---

### Public Comment · (anonymous) · 2017-11-21
**The structure of TreeQN and how to train the model functions**

The idea that integrate model planning into the Q-function or policy is interesting.

However, I wander how the model functions (transition function, reward function and value function) are trained. From the description of the paper, they may be special designed sub-networks, the whole network is trained based on the TD-error of n-step Q-learning. If it is, how can we know the sub-networks are planning indeed?

In my opinion, TreeQN is a complex network with a fixed planning step (i.e., the tree depth). And each planning step is a special designed sub-network. The sub-network is similar as CNN, which can extract useful feature from the state representation z.

Do I understand this correctly?

---

> ### Author Response · Authors · 2017-11-22
> **An inductive bias that encourages planning**
>
> Thank you for your comments! It is true that the whole network is trained end-to-end based on the TD-error of n-step Q-learning and so there is no guarantee that the trained sub-networks learn a faithful model of the environment (i.e., the internal state representations are not guaranteed or necessarily expected to contain the information needed to accurately predict observations in future time-steps).
>
> However, this flexibility is intentional because, at the end of the day, we only care about predicting accurate state-action values. Consequently, we want to make our architecture flexible enough to learn an abstract representation of the environment and a transition model that, when used together inside TreeQN, are effective at predicting those state-action values even if the resulting architecture does not correspond to rigid definitions of model, state, or plan. Instead, we incorporate an inductive bias through the recursive application of a learned transition function that is shared across the tree. This inductive bias is introduced through the network’s structure and does indeed encourage planning.
>
> Note that there is a weak grounding of predicted states via our reward prediction auxiliary loss. Furthermore, our results show that the model does produce interpretable trees in some situations, which demonstrates that grounded planning is performed when useful, even though the model is not limited to it. We experimented with ways of further grounding the transition function (and thus the states) but found that it only hurt performance.  Finding ways to encourage stronger grounding without hurting performance is an interesting research direction, but not in the scope of this paper.
>
> The sub-networks (transition, reward and value function) are not CNNs (see Equations 6 to 8).

---

> > ### Public Comment · (anonymous) · 2017-11-23
> > **Although the inductive bias cannot learn the exact models of the environment, I appreciate this method very much.**
> >
> > I understand this method now. However, if add the comments below in the original paper, readers can understand it more easily. Overall, this method is very interesting and insightful, I appreciate it very much.
> >
> >  ''There is no guarantee that the trained sub-networks learn a faithful model of the environment''
> > ''However, this flexibility is intentional because, at the end of the day, we only care about predicting accurate state-action values. Consequently, we want to make our architecture flexible enough to learn an abstract representation of the environment and a transition model that, when used together inside TreeQN, are effective at predicting those state-action values even if the resulting architecture does not correspond to rigid definitions of model, state, or plan.''
> >
> >
> > The sub-networks are not CNNs indeed. I want to express that the sub-networks can mimic the planning, while CNN can extract local features, and both of them are universal modules although designed for special purposes.

---

### Public Comment · (anonymous) · 2017-12-28
**Interesting and an easy followed paper, yet some details are missing**

Our group reproduced this paper and the detailed result is in the link.

https://gitlab.eecs.umich.edu/kongfz/TreeQN_ATreeC_repro/blob/master/report.pdf

Hope our feedback can improve your paper!

---

> ### Author Response · Authors · 2018-01-04
> **Thank you for your feedback and interest; several bugs in the re-implementations.**
>
> Thank you very much. We really appreciate your time and effort in trying to reproduce our results; this is a great initiative. We read through your report and online code, and identified several issues and important bugs that we list and discuss below. We intend to open source our code in time (after the anonymous review period), but are happy to correspond further in advance of that, if you’re interested in attempting further to reproduce our results.
>
> Issues in the report:
>     - All results on the Box pushing environment seem to be off. The average reward is negative, even for the DQN baseline. This suggests there is something fundamentally different in your implementation.
>     - Cartpole is not a suitable test environment. It is a very simple environment and there is no reason to believe that ad-hoc planning is required or helps. Indeed, any reasonably tuned algorithm should reach a performance ceiling on this toy task, so there is no reason TreeQN should outperform DQN.
>     - Appendix: “unsure which part of the network is minimized” -- we aren’t sure where the misunderstanding lies (whole network is trained end-to-end and overall loss is minimized)
>     - In our paper “DQN” refers to the architecture rather than the training algorithm, which is n-step Q-learning as described in the background section -- we do not use a replay buffer. However, we hope and expect that a proper TreeQN implementation would help with off-policy training data as well, and it is worth investigation.
>
> Errors in code:
>     TreeQN (Tensorflow implementation)
>         - The Q-learning algorithm implementation for TreeQN looks wrong. You need to compute targets using the target network evaluation of next_states. In line 74 of TreeQNModel.py these targets are computed using the current state, and do not include the observed reward.
>         - Action selection should use the online network rather than the target network.
>     ATreeC (Pytorch implementation)
>         - For the CNN encoder (our encoder) there doesn’t appear to be any residual connection or nonlinearity (nonlinearities between layers are crucial!) in the transition function.
>         - The value module should be a different set of parameters from the critic’s fully-connected layer, as shown in fig. 3 of our paper. We can make this more clear in the paper.
>         - There should be an auxiliary loss on the reward prediction; we use this for both TreeQN and ATreeC.
>         - Don’t subtract the critic value from Q estimates; critic heads are not used in policy at all. The “Q-estimates” from the tree should go straight into a softmax to produce the policy.
>         - For your fully-connected variant of ATreeC’s encoder, try using the residual connection just for transition function; don’t add again the original inputs from before the encoder layers.
>         - NB: there may be a more fundamental issue, as the failure of any algorithm to learn in fig. 3 of the report is surprising.
>
> Hyperparameters:
>     - We will update the paper to include: discount factor gamma=0.99, and the target networks are updated every 40000 environment transitions; for both box-pushing and Atari.

---

### Decision · Program_Chairs · 2018-01-29
**ICLR 2018 Conference Acceptance Decision**

**Decision:**

Accept (Poster)

**Comment:**

This is a nicely written paper proposing a reasonably interesting extension to existing work (e.g.
VPN). While the Atari results are not particular convincing, they do show promise. I encourage
the authors to carefully take the reviewers' comment into consideration and incorporate them
to the final version.